# Littlestone Classes are Privately Online Learnable

**Noah Golowich**
MIT CSAIl
nzg@mit.edu

**Roi Livni**
Tel Aviv University
rlivni@tauex.tau.ac.il

## Abstract

We consider the problem of online classification under a privacy constraint. In this setting a learner observes sequentially a stream of labelled examples $(x_t, y_t)$, for $1 \le t \le T$, and returns at each iteration $t$ a hypothesis $h_t$ which is used to predict the label of each new example $x_t$. The learner's performance is measured by her regret against a known hypothesis class $\mathcal{H}$. We require that the algorithm satisfies the following privacy constraint: the sequence $h_1, \ldots, h_T$ of hypotheses output by the algorithm needs to be an $(\epsilon, \delta)$-differentially private function of the whole input sequence $(x_1, y_1), \ldots, (x_T, y_T)$. We provide the first non-trivial regret bound for the realizable setting. Specifically, we show that if the class $\mathcal{H}$ has constant Littlestone dimension then, given an oblivious sequence of labelled examples, there is a private learner that makes in expectation at most $O(\log T)$ mistakes – comparable to the optimal mistake bound in the non-private case, up to a logarithmic factor. Moreover, for general values of the Littlestone dimension $d$, the same mistake bound holds but with a doubly-exponential in $d$ factor. A recent line of work has demonstrated a strong connection between classes that are online learnable and those that are differentially-private learnable. Our results strengthen this connection and show that an online learning algorithm can in fact be directly privatized (in the realizable setting). We also discuss an adaptive setting and provide a sublinear regret bound of $O(\sqrt{T})$.

## 1   Introduction

Privacy-preserving machine learning has attracted considerable attention in recent years, motivated by the fact that individuals' data is often collected to train statistical models, and such models can leak sensitive data about those individuals [13, 32]. The notion of *differential privacy* has emerged as a central tool which can be used to formally reason about the privacy-accuracy tradeoffs one must make in the process of analyzing and learning from data. A considerable body of literature on *differentially private machine learning* has resulted, ranging from empirical works which train deep neural networks with a differentially private form of stochastic gradient descent [1], to a recent line of theoretical works which aim to characterize the optimal sample complexity of privately learning an arbitrary hypothesis class [3, 11, 20].

Nearly all of these prior works on differentially private learning, however, are limited to the statistical learning setting (also known as the *offline* setting): this is the setting where the labeled data, $(x_t, y_t)$, are assumed to be drawn i.i.d. from some unknown population distribution. This setting, while very well-understod and readily amenable to analysis, is unlikely to hold in practice. Indeed, the data $(x_t, y_t)$ fed as input into the learning algorithm may shift over time (e.g., as a consequence of demographic changes in a population), or may be subject to more drastic changes which are *adaptive* to the algorithm's prior predictions (e.g., drivers' reactions to the recommendations of route-planning apps may affect traffic patterns, which influence the input data to those apps). For this reason, it is desirable to develop provable algorithms which make fewer assumptions on the data.

35th Conference on Neural Information Processing Systems (NeurIPS 2021).

In this work, we do so by studying the setting of *(private) online learning*, in which the sequence of data $(x_t, y_t)$ is allowed to be arbitrary, and we also discuss a certain notion of privacy in a setting where it is even allowed to adapt to the algorithm's predictions in prior rounds. We additionally restrict our attention to the problem of classification, namely where the labels $y_t \in \{0, 1\}$; thus we introduce the problem of differentially private online classification, and prove the following results (see Section 3 for the exact setup):

- In the realizable setting with an oblivious adversary, we introduce a private learning algorithm which, for hypothesis classes of Littlestone dimension $d$ (see Section 2.1) and time horizon $T$, achieves a mistake bound of $\tilde{O}(2^{O(2^d)} \cdot \log T)$, ignoring the dependence on privacy parameters (Theorem 4.1).

- In the realizable setting with an adaptive adversary, we show that a slight modification of the above algorithm achieves a mistake bound of $\tilde{O}(2^{O(2^d)} \cdot \sqrt{T})$ (Theorem 4.2).

We remark that no algorithm (even without privacy, allowing randomization, and in the oblivious adversary setting) can achieve a mistake bound of smaller than $\Omega(d)$ for classes of Littlestone dimension $d$ [30, 33]. Therefore, a class of infinite Littlestone dimension cannot have any finite mistake bound, and the regret for any algorithm, for any time horizon $T$, is $\Omega(T)$. Thus, our results listed above, which show a mistake-bound (which is also the regret in the realizable setting) of $\tilde{O}_d(\sqrt{T})$ for classes of Littlestone dimension $d$, establish that in the realizable setting, finiteness of the Littlestone dimension is necessary and sufficient for online learnability ([31]) with differential privacy.

Recently it was shown by Alon et al. [3] and Bun et al. [11] (later to be improved by Ghazi et al. [20]) that finiteness of the Littlestone dimension is necessary and sufficient for private learnability in the *offline* setting, namely with i.i.d. data (and both in the realizable and agnostic settings). Since, as remarked above, the Littlestone dimension characterizes online learnability (even without privacy), this means that a binary hypothesis class is privately (offline) learnable if and only if it is online learnable. Our result thus strengthens this connection, showing that the equivalence also includes private *online* learnability (in the realizable setting).

## 1.1 Related work

A series of papers [15, 25, 21, 17, 2] has studied the problem of diferentially private online convex optimization, which includes specific cases such as private prediction from expert advice and, when one assumes imperfect feedback, private non-stochastic multi-armed bandits [35, 36, 18, 24]. These results show that in many regimes privacy is free for such problems: for instance, for the problem of prediction from the expert advice (with $N$ experts), Agarwal and Singh [2] shows that an $\epsilon$-differentially private algorithm (based on follow-the-regularized-leader) achieves regret of $O\left(\sqrt{T} + \frac{N \log^2 T}{\epsilon}\right)$, which matches the non-private regret bound of $O(\sqrt{T \log N})$ when $T \geq \tilde{\Omega}((N/\epsilon)^2)$. Our results can be seen as extending such "privacy is (nearly) free" results to the nonparametric setting where we instead optimize over an arbitrary class of finite Littlestone dimension. Our techniques are different from those of the above papers.

In addition to [11, 20] which establish private learning algorithms for classes with finite Littlestone dimension in the i.i.d. (offline) setting, there has been an extensive line of work on private learning algorithms in the offline setting: [29, 7, 5, 19] study the complexity of private learning with *pure differential privacy*, [26, 9, 10, 4] study the sample complexity of privately learning thresholds, and [27, 28, 6] study the sample complexity of privately learning halfspaces.

## 2 Preliminaries

In this section we introduce some background concepts used in the paper.

## 2.1 Online Learning

We begin by revisiting the standard setting of online-learning: We consider a sequential game between a *learner* and an *adversary*. Both learner and adversary know the sets $\mathcal{X}$ and $\mathcal{H}$. The game

proceeds for $T$ rounds (again $T$ is known) and at each round $t \leq T$, the adversary chooses a pair $(x_t, y_t)$ and presents the learner with the example $x_t$. The learner then must present the adversary with a hypothesis (perhaps randomly) $h_t : \mathcal{X} \to \{0, 1\}$. $h_t$ is not required to lie in $\mathcal{H}$[1]. Finally the adversary presents the learner with $y_t$, which the learner uses to update its internal state. The performance of the learner is measured by its regret which is its number of mistake vs. the optimal decision in hindsight:

$$\mathbb{E}\left[ \sum_{t=1}^{T} 1[h_t(x_t) \neq y_t] - \min_{h^\star \in \mathcal{H}} \sum_{t=1}^{T} 1[h^\star(x_t) \neq y_t] \right]. \tag{1}$$

The adversary is said to be *realizable* if it presents the learner with a sequence of examples $(x_t, y_t)$ so that there is some $h^\star \in \mathcal{H}$ so that for each $t \in [T]$, $h^\star(x_t) = y_t$. In the realizable setting, the regret simply counts the number of mistakes the learner makes. And we measure the performance by its *mistake bound*, namely the maximum, over all possible realizable adversaries, of

$$\mathbb{E}\left[ \sum_{t=1}^{T} 1[h_t(x_t) \neq y_t] \right].$$

In the setting with an *agnostic* adversary, we do not require such $h^\star$ to exist; and we measure the learner by its (worst-case) *regret*, as in Eq. (1). In this paper we focus on the realizable setting; the (private ) agnostic setting is left as an interesting direction for future work.

Additionally, we normally make a distinction between two types of adversaries: An *oblivious* adversary chooses its sequence in advance and at each iteration $(x_t, y_t)$ is revealed to the learner. In the *adversarial* setting, the adversary may choose $(x_t, y_t)$ as a function of the learner's previous choices: i.e. $h_1, \ldots, h_{t-1}$. This definition follows the standard setup of online learning (see [12] for example). We note though, that in the non-private setting of online binary classification, one can obtain results against an adversary that even gets to observe the learner's prediction at time-step $t$. However, we will simplify here by considering the more standard setting. It is interesting to find out if we can compete against such a strong adversary in the private setup.

**Littlestone dimension** We next turn to introduce the Littlestone dimension which is a combinatorial measure that turns out to characterize learnability in the above setting.

Let $\mathcal{H}$ be a class of hypotheses $h : \mathcal{X} \to \{0, 1\}$. To define the Littlestone dimension of $\mathcal{H}$, we first introduce *mistake trees*: a mistake tree of *depth $d$* is a complete binary tree, each of whose non-leaf nodes $v$ is labeled by a point $x_v \in \mathcal{X}$, and so that the two out-edges of $v$ are labeled by 0 and 1. We associate each root-to-leaf path in a mistake tree with a sequence $(x_1, y_1), \ldots, (x_d, y_d)$, where for each $i \in [d]$, the $i$th node in the path is labeled $x_i$ and the path takes the out-edge from that node labeled $y_i$. A mistake tree is said to be *shattered* by $\mathcal{H}$ if for any root-to-leaf path whose corresponding sequence is $(x_1, y_1), \ldots, (x_d, y_d)$, there is some $h \in \mathcal{H}$ so that $h(x_i) = y_i$ for all $i \in [d]$. The Littlestone dimension of $\mathcal{H}$, denoted $\mathrm{Ldim}(\mathcal{H})$, is the depth of the largest mistake tree that is shattered by $\mathcal{H}$.

**The Standard Optimal Algorithm (SOA)** Suppose $\mathcal{H}$ is a binary hypothesis class with Littlestone dimension $d$. Littlestone [30] showed that there is an algorithm, called the *Standard Optimal Algorithm* (SOA), which, against an adaptive and realizable adversary, has a mistake bound of $d$; moreover, this is the best possible mistake bound. We will access the SOA as a black box. The underlying assumption we make is that given a realizable sequence $(x_1, y_1), \ldots, (x_T, y_T)$, the $SOA$ makes at most $\mathrm{Ldim}(\mathcal{H})$ mistakes. We will also assume that whenever the algorithm $SOA$ makes a mistake then it changes it state: namely, if the algorithm makes mistake on example $t$ then $h_{t+1} \neq h_t$, this is in fact true for the SOA algorithm, but it can be seen that any algorithm with mistake bound can be modified to make sure this holds (simply by reiterating the mistake until the algorithm does change state). We refer the reader to [30, 33] for the specifics of it.

## 2.2 Differential Privacy

We next recall the standard notion of $(\epsilon, \delta)$–differential privacy:

---

[1]This setup is known as the *improper* learning problem. In the *proper* version of the problem, it is required that $h_t \in \mathcal{H}$ and we leave a study of proper private online learning for future work. (see [22] for a discssion on proper online learning in the non-private case

**Definition 2.1** (Differential privacy). Let $n$ be a positive integer, $\epsilon, \delta \in (0,1)$, and $\mathcal{W}$ be a set. A randomized algorithm $A : (\mathcal{X} \times \{0,1\})^n \to \mathcal{W}$ is defined to be $(\epsilon, \delta)$-*differentially private* if for any two datasets $S, S' \in (\mathcal{X} \times \{0,1\})^n$ differing in a single example, and any event $\mathcal{E} \subset \mathcal{E}$, it holds that

$$\Pr[A(S) \in \mathcal{E}] \le e^\epsilon \cdot \Pr[A(S') \in \mathcal{E}] + \delta.$$

**Adaptive Composition** The online nature of the problem naturally requires us to deal with adaptive mechanisms that query the data-base. We thus depict here the standard framework of adaptive querying, and we refer the reader to Dwork and Roth [13] for a more detailed exposition.

In this framework we assume a sequential setting, where at step $t$ an adversary chooses two adjacent datasets $S_t^1$ and $S_t^0$, and a mechanism $M_t(S)$ from a class $\mathcal{F}$ and receives $z_t^b = M_t(S_t^b)$ for some $b \in \{0,1\}$ (where $b$ does not depend on $t$).

**Definition 2.2.** We say that the family $\mathcal{F}$ of algorithms over databases satisfies $(\epsilon, \delta)$-*differential privacy under $T$-fold adaptive composition* if for every adversary $A$ and event $\mathcal{E}$, we have

$$\Pr((z_1^0, \ldots, z_T^0) \in \mathcal{E}) \le e^\epsilon \Pr((z_1^1, \ldots, z_T^1) \in \mathcal{E}) + \delta.$$

# 3 Problem Setup

We now formally introduce the main problem considered in this paper, namely that of *private online learning*. Let $\mathcal{X}$ be a set, and let $\mathcal{H}$ be a set of *hypotheses*, namely of functions $h : \mathcal{X} \to \{0,1\}$. We consider the setting depicted in Section 2.1 and in this framework we want to study the learnability of *private* learners which are defined next. We make a distinction between the case of an oblivious and an adaptive adversary:

**Private online learning vs. an *oblivious* adversary** As discussed, in this setting the adversary must choose the entire sequence $(x_1, y_1), \ldots, (x_T, y_T)$ before its interaction with the learner (though it may use knowledge of the learner's *algorithm*). In particular, the samples $(x_t, y_t)$ do not depend on any random bits used by the learner. Thus, in the *private online learning problem* we merely require that the sequence of hypotheses $(h_1, \ldots, h_T)$ output by the learner is $(\epsilon, \delta)$-differentially private as a function of the entire input sequence $(x_1, y_1), \ldots, (x_T, y_T)$.

**Private online learning vs. an *adaptive* adversary:** In the adaptive setting, the adversary may choose each example $(x_t, y_t)$ as a function of all of the learner's hypotheses up to $t$. This makes the notion of privacy a little bit more subtle, so we need to carefully define what we mean here by $(\epsilon, \delta)$-privacy. We consider then the following scenario:

At each round $t$, the adversary outputs two outcomes $(x_t^0, y_t^0)$ and $(x_t^1, y_t^1)$. The learner then outputs $h_t^b$ and $(x_t^b, y_t^b)$ is revealed to the learner where $b \in \{0,1\}$ is independent of $t$. We require that the sequences $S_T^0 = \{(x_t^0, y_t^0)\}$ and $S_T^1 = \{(x_t^1, y_t^1)\}$ differ in, at most, a single example. We will say that an adaptive online classification algorithm is $(\epsilon, \delta)$ differentially private, if for any event $\mathcal{E}$ and any adversary, it holds that

$$\Pr[(h_1^1, \ldots, h_T^1) \in \mathcal{E}] \le e^\epsilon \cdot \Pr[(h_1^0, \ldots, h_T^0) \in \mathcal{E}] + \delta.$$

The notion is similar to privacy under $T$-fold adaptive composition. Normally, though, for a mechanism to be $(\epsilon, \delta)$-differentially private under $T$-fold adaptive compositions, Dwork et al. [16] requires it to be private under an adversary that may choose at each iteration *any* two adjacent datasets, $S_i^0$, $S_i^1$. Note, however that, in the online setup, the utility is dependent only on a single point at each iteration, hence such a requirement will be too strong (in fact, the learner will then be tested on two arbitrary sequences).

# 4 Main Results

We next state the main results of this paper, we start with a logarithmic regret bound for realizable oblivious learning.

**Theorem 4.1** (Private Oblivious online-learning). *For a choice of $k_1 = \tilde{O}(2^{d+1})$, and*

$$k_2 = \tilde{O}\left(\frac{2^{8 \cdot 2^d}}{\epsilon} \ln T/\delta\right),$$

*Running* DP-SOA *(Algorithm 1) for T iterations on any realizable sequence* $(x_1, y_1), \ldots, (x_T, y_T)$, *the algorithm outputs a sequence of predictors* $h_1, \ldots, h_T$ *such that*

- *The algorithm is* $(\epsilon, \delta)$ *differentially private.*

- *The expected number of mistakes the algorithm makes is*

$$\mathbb{E}[\sum_{t=1}^{T} h_t(x_t) \neq y_t] = \tilde{O}\left(\frac{2^{8 \cdot 2^d}}{\epsilon} \ln T / \delta\right).$$

Theorem 4.1 shows that, up to logarithmic factor, the number of mistakes in the private case is comparable with the number of mistakes in the non-private case, when $d$ the Littlestone dimension of the class is constant. We obtain, though, a strong deterioration in terms of the Littlestone dimension – sublinear dependece vs. double exponential dependence. As discussed, Ghazi et al. [20] improved the dependence in the batch case to polynomial, and it remains an open question if similar improvement is applicable in the online case. We next turn to the adversarial case

**Theorem 4.2** (Private Adaptive online-learning)**.** *There exists an adaptive online classification algorithm that is* $(\epsilon, \delta)$-*differentially private with expected regret over a realizble seqeunce:*

$$\mathbb{E}[\sum_{t=1}^{T} h_t(x_t) \neq y_t] = \tilde{O}\left(\frac{2^{O(2^d)}\sqrt{T}\log 1/(\delta)}{\epsilon}\right).$$

Theorem 4.2 provides a sublinear regret bound, which is in fact optimal for the agnostic case. However, in the non-private (realizable) case it is known that constant regret can be obtained[2] . We leave it as an open problem whether one can achieve logarithmic regret in the realizable adaptive setting.

## 5 Algorithm

We next present our main algorithm for an oblivious, realizable online private learning algorithm. The algorithm, DP-SOA, assumes access to a mistake bound algorithm for the class $\mathcal{H}$ (not necessarily private) such as SOA as in [30], which we denote by $A$,[3] as well as call a procedure HistSparse that is depicted below (Algorithm 2). We can think of DP-SOA as an algorithm that runs several copies of the same procedure, where each copy is working on its own subsequence of $(x_1, y_1), \ldots, (x_T, y_T)$, and the sub sequences form a random partition of the entire sequence.

Each process can be described by a tree whose vertices are labelled by samples that are iteratively constructed. Each tree outputs a predictor according to the state of its vertices. Hence, overall the algorithm can be depicted as a forest, where at each iteration an example is randomly assigned to one of the trees, and that tree, in turn, makes an update.

At each time step, we maintain a set of vertices $\mathcal{V}_t$, which we will call *pertinent* vertices. Each pertinent vertex $v$ holds a sample $S_v$. At time $t = 1$ only the leaves are in $\mathcal{V}_1$, and each leaf $v$ is assigned the sample $S_v = \emptyset$. Then, at every time-step where an example $(x_t, y_t)$ is assigned to the tree, it is randomly assigned to a pertinent vertex $v$ in $\mathcal{V}$ (in detail, it is first randomly assigned to a leaf and then propagated to a pertinent ancestor), and the sample $S_v$ is updated to $(S_v, (x_t, y_t))$. After that, as we next describe, a process starts that updates the set of pertinent vertices; this process follows the idea of the tournament examples presented in [11].

Whenever two siblings $v, s(v)$ are pertinent and assigned with sequences $S_v$ and $S_{s(v)}$, respectively, they stay pertinent as long as $A(S_v) = A(S_{s(v)})$, and samples are assigned to them at their turn via the process depicted above. Whenever it becomes the case that $A(S_v) \neq A(S_{s(v)})$, let $\bar{v}$ denote the parent of $v, s(v)$; we consider an example $x_{\bar{v}}$ on which $A(S_v)$, $A(S_{s(v)})$ disagree, and guess its label $y_{\bar{v}}$. Then, $v, s(v)$ are removed from the set of pertinent vertices, their parent $\bar{v}$ becomes pertinent, and we set $S_{\bar{v}}$ to equal $(S_v, (x_v, y_v))$ if $A(S_v)[x_v] \neq y_v$, and $(S_{s(v)}, (x_v, y_v))$ otherwise. Once this

---

[2] and as discussed, the adversary may even depend on $h_t$ at round $t$

[3] In particular, $A$ is required to be an algorithm that achieves a mistake bound of at most $d$ on hypothesis classes of Littlestone dimension $d$. We will use the following (easily verified) fact about such an algorithm: after making a mistake, the algorithm must change the hypothesis it outputs for the following round.

---
**Algorithm 1** DP-SOA
---
Input $(\epsilon, \delta)$, $k_1, k_2$.
Set $\eta = \frac{2^{-4k_1}}{4k_1}$, and $c = 4k_1/\eta$
Let $G = (V, E)$ be a forest of $k_2$ full binary trees, each with $k_1$ leaves.
Let $\pi : T \to \text{Leaves}(V)$ be a random mapping that maps $t \in [T]$ to a random leaf.
Set $S_v = \emptyset$ for each leaf $v$ and $S_u = \perp$ for each non-leaf vertex $u$ (where we define $A(\perp) = \perp$).
Initialize $\mathcal{V}_1$ to be the set of all leaves in the forest.
set $v_1^{(i)}$ be an arbitrary leaf from the tree $G_i$, for each $i \in [k_2]$
**for** t=1 to T **do**
    Run $HistSparse_{\epsilon, \delta, \eta, c}(h_{t-1}, L_t)$ on the List $L_t = \{A(S_{v_t^{(i)}})\}_{i=1}^{k_2}$ and receive $h_t$
    Predict $h_t(x_t) = \hat{y}_t$, and observe $y_t$.
    Choose $v_1 \in \mathcal{V}_t$ to be an antecedent of leaf $\pi(t)$ %there exists a unique antecedent in $\mathcal{V}_t$
    Set $v_2 = s(v_1)$ (if $v_1$ is the root, continue to the next iteration).
    Set $(S_{v_1}, (x_t, y_t)) \to S_{v_1}$.
    **while** $A(S_{v_1}) \neq A(S_{v_2})$ AND $v_1, v_2 \in \mathcal{V}_t$ **do**
        Set $\bar{v}$ to be the parent of $v_1, v_2$
        Choose an arbitrary $x_{\bar{v}}$ such that $A(S_{v_1})[x_{\bar{v}}] \neq A(S_{v_2})[x_{\bar{v}}]$ and $y_{\bar{v}}$ randomly
        Set $(S_{v_i}, (x_{\bar{v}}, y_{\bar{v}})) \to S_{\bar{v}}$ where $i$ is such that $A(S_{v_i})[x_v] \neq y_v$.
        Remove $v_1, v_2$ from $\mathcal{V}_t$ and add $\bar{v}$ to $\mathcal{V}_t$.
        Let $v_1$ be $\bar{v}_t$
        **if** $v_1$ is not the root **then**
            Set $v_2$ to be the sibling of $v_1$
        **else**
            Set $v_1 = v_2$ (and hence exit the loop.)
        **end if**
    **end while**
    **if** The While loop was executed at least once **then**
        Let $i$ be the tree for which $\pi(t)$ belongs to.
        Choose randomly a vertex $v$ in tree $i$ such that $v, s(v) \in \mathcal{V}_t$ and $A(S_v) = A(S_{s(v)})$ (break ties by choosing randomly).
        (If no such $v$ exists, let $v$ be the root and set $S_v$ to be some sample for which $A(S_v) = \perp$, add the root to $\mathcal{V}_t$ and remove all other vertices that belong to tree $i$).
        Set $v_{t+1}^{(i')} = \begin{cases} v & i = i' \\ v_t^{(i')} & i \neq i' \end{cases}$.
    **else**
        Set $v_{t+1}^{(i')} = v_t^{(i')}$ for all $i' \leq k_2$.
    **end if**
    Set $\mathcal{V}_{t+1} = \mathcal{V}_t$.
**end for**
---

procedure finishes, the tree outputs (randomly) some hypothesis $h = A(S_v)$ where $v$ is a pertinent vertex. The hypothesis will change only when the state of the tree changes (note that at initialization, the tree outputs $A(\emptyset)$).

## 5.1 Technical Overview

We next give a high level overview of our proof techniques. We focus until the end of this section on the oblivious realizable case. The main procedure of the algorithm, DP-SOA, is Algorithm 1.

Our proof strategy is similar to the approach of Bun et al. [11] for learning privately in the stochastic setting, which we next briefly describe. In the stochastic setup, the idea was to rely on *global stability*. In a nutshell, a randomized algorithm is called globally stable if it outputs a certain function with constant probability (over the random bits of the algorithm as well as the random i.i.d sample). Once we can construct such an algorithm (with sufficiently small error) we run several copies of the algorithm on separate samples, and then we can use any mechanism, such as the one in Theorem 5.1 below, that publishes (privately) an estimated histogram of the frequency of appearance of each

---

**Algorithm 2** `HistSparse`: Receives a sequence of 1-sensitive lists $L_1(D), \ldots, L_T(D)$.

---

   **Initialize:** parameters $\epsilon, \eta, \delta, c$.
   Let $\sigma = 2c/(k\epsilon)$, $\theta = 1 - 3\eta/32$
   Let $\theta_0 = \theta + \text{LAP}(\sigma)$.
   Let counter $= 1$
   For list $L_1$ set $h_1 = hist_{\epsilon/(2c,\delta/c,\eta)}(L_1)$.
   **for** $t = 1, \ldots, T$: **do**
      Define query: $Q_t = 1 - \text{freq}_{L_t}(h_{t-1})$.
      Let $v_i = \text{LAP}(2\sigma)$
      **if** $Q_t + v_i \geq \theta_{\text{counter}}$ **then**
         Set $h_t = hist_{\epsilon/(2c),\delta/c,\eta}(L_t)$
         counter $=$ counter $+ 1$
         $\theta_{\text{counter}} = \theta + \text{LAP}(\sigma)$.
      **else**
         Set $h_t = h_{t-1}$
      **end if**
      **if** counter $\geq c$ **then**
         ABORT
      **end if**
   **end for**

---

function. In detail, given a list $L = \{x_1, \ldots, x_k\}$ we denote by $\text{freq}_L$ the mapping

$$\text{freq}_L(f) = \frac{1}{k} \sum_{x \in L} \mathbf{1}[x = f].$$

**Theorem 5.1** ([8] essentially Proposition 2.20). *For every $\epsilon, \delta$ and $\eta$, there exists a $(\epsilon, \delta)$-DP mechanism $hist_{\epsilon,\delta,\eta}$ that given a list $L = \{x_1, \ldots, x_k\}$, outputs a mapping $\overline{\text{freq}}_L : \mathcal{X} \to [0, 1]$ such that if*

$$k \geq \Theta_{(2)}(\eta, \beta, \epsilon, \delta) := 4/\eta + \frac{\log 1/(\eta^2 \beta \delta)}{\eta \epsilon} = O\left(\frac{\log 1/\eta \beta \delta}{\eta \epsilon}\right), \tag{2}$$

*then with probability $(1 - \beta)$:*

- *If $\overline{\text{freq}}_L(x) > 0$ then $\text{freq}_L(x) > \frac{\eta}{4}$.*

- *For every $x$ such that $\text{freq}_L(x) > \eta$, we have that $\overline{\text{freq}}_L(x) > 0$.*

Our algorithm follows a similar strategy but certain care needs to taken due to the sequential (and distribution-free) nature of the data, as well as the fact that using `hist` procedure $T$ times may be prohibitive (if we wish to obtain logarithmic regret). We next review these challenges:

**Global Stability** Our first task is to construct an online version of a globally stable algorithm, which roughly means that different copies of the same algorithm run on disjoint subsequences of $(x_1, y_1), \ldots, (x_T, y_T)$, and output a fixed hypothesis which may depend on the whole sequence but not on the disjoint subsequences. `DP-SOA` does so by assigning each subsequence to a tree which is running the procedure described in Section 5. We now explain how this procedure induces the desired stability.

As in Section 5, recall that a vertex $v$ is pertinent if it is in the set $\mathcal{V}_t$. We will refer to the distance of a vertex to any of its leaves as that vertex's *depth*. Note that for each pertinent vertex $v$ at depth $k$, the algorithm makes $k$ mistakes on the sequence $S_v$ – indeed, whenever a vertex $\bar{v}$ is made pertinent, we always append to $S_{\bar{v}}$ an example which forces a mistake for the sequence of a child of $\bar{v}$. Also, notice that with probability $2^{-2k_1}$, where $k_1$ is the number of leaves in the tree, all sequences assigned to each pertinent vertex are consistent with the realized hypothesis $h^\star$ (recall that we are considering here the oblivious realizable case, hence $h^\star$ is well-defined). Indeed, this is true as as long as we guessed the label $y_{\bar{v}}$ to equal $h^\star(x_{\bar{v}})$ at each round; the number of guesses is bounded by the number of vertices, which is $2k_1 - 1 < 2k_1$. Ultimately, this allows two cases: in the first case a vertex of depth $d$ is pertinent: in this case the vertex must identify $h^\star$ (indeed, if there are two different

hypotheses that are consistent on a sample with $d$ mistakes, then we can force a $(d + 1)$th mistake). So, if there are "many" trees with a $d$-depth pertinent vertex, then fraction of $2^{-2k_1}$ of them, are outputting $h^\star$, hence we found a frequent hypothesis. The second case is that in "many" of the trees, for some $k < d$, there are many pairs $v, s(v)$ of pertinent vertices at depth $k$ so that $A(S_v) = A(S_{s(v)})$; we will refer to such a pair $v, s(v)$ as a *collision*.

In the batch case the latter case immediately implies that some hypothesis is outputted frequently (i.e., we get global stability) through a standard concentration inequality that relates the number of collisions between i.i.d random variables, and the frequency of the most probable hypothesis. In the online case it is a little bit more subtle as the examples are not i.i.d, hence the sequences for the pertinent vertices are not i.i.d copies of some random variable. However, suppose that there are many collisions at depth $k$, and that we now reassign the data by randomly permuting the $k$-depth subtree (i.e. we reassign a random parent to each vertex at depth $k$, in order to form a new complete binary tree, and we don't change relations at other depths). Since the assignment of the data $(x_t, y_t)$ to the leaves is invariant under permutation, we can think of this process as randomly picking a new assignment, conditioning on the $k$-th level structure of the trees. Alternatively, we can also think of this process as randomly picking without replacement the different hypotheses outputed by the $k$-depth vertices, and counting collisions of siblings.

We now want to relate the number of collisions to their expected mean and obtain a bound on the most frequent hypothesis. We can do this using a variant of Mcdiarmid's inequality for permutations – or sampling without replacement. The observation for this inequality was found in [23] which attributes it to Talagrand [34]. For completeness we provide the proof in the full version.

**Lemma 5.2** (Mcdiarmid's without replacement). *Suppose $\bar{Z} = (Z_1, \ldots, Z_n)$ are random variables sampled uniformly from some universe $\mathcal{Z} = \{z^{(1)}, \ldots, z^{(N)}\}$ without replacement (in particular $n \le N$). Let $F : Z^n \to [0, 1]$ be a mapping such that for $\bar{z} = (z_1, \ldots, z_n)$ and $\bar{z}' = (z'_1, \ldots, z'_n)$ that are of Hamming distance at most 1, $|F(\bar{z}) - F(\bar{z}')| \le c$. Then:*

$$\mathbb{P}\left(\mathbb{E}(F(\bar{Z})) - F(\bar{Z}) \ge \epsilon\right) \le e^{-\frac{2\epsilon^2}{9nc^2}}.$$

We use Lemma 5.2 as follows: our function $F$ counts the number of collisions between depth $k$ vertices after a random permutation (where we think here of permutation as sampling without replacement), this function is 1-sensitive to changing a single element, as required. We thus obtain an estimate of the number of collisions for a random permutation, which we can relate to the appearance of the most frequent hypothesis.

The above calculation can be used to obtain a guarantee that there exists an hypothesis that appears at frequency $2^{-O(k_1)}$ (this frequency is roughly the probability that the tree remains consistent with $h^\star$). Since the number of leaves is exponential in the depth, and the depth needs to be at least $d$ (the upper bound on the level at which the algorithm stabilizes for sure), we overall obtain doubly exponential dependence of the frequency on the Littlestone dimension.

**Mistake Bound** We next turn to bound the number of mistakes. The crucial observation is that every time the algorithm makes a mistake, if example $x_t$ is assigned to tree $i$ then with some positive probability (specifically, the frequency of $h_t$, lower bounded by $2^{-O(2^d)}$) tree $i$ outputs $h_t$. Moreover, with probability $1/k_1 > 0$, $x_t$ is assigned to the pertinent vertex that made the mistake. Once the example is assigned to this vertex, we have $A((S_v, (x_t, y_t))) \ne A(S_{s(v)})$. In particular, the two siblings are taken out of the list of pertinent vertices, and their parent becomes pertinent. In other words, every time the algorithm makes a mistake with some constant probability (roughly $2^{-\tilde{O}(2^d)}$), the set of pertinent vertices diminishes by one. Since we start with finite number of leaves as pertinent vertices, the expected number of mistakes is bounded by the number of leaves in the forest.

It remains to show that the number of leaves in the forest is logarithmic in the sequence size (but doubly exponential in the Littlestone dimension). The number of leaves is roughly $k_1$ (which is roughly $O(2^d)$) times the number of trees in the forest; this number of trees depends on the sample complexity of the private process in which we output the frequent hypothesis. We now explain why roughly $O(2^{O(2^d)} \ln T)$ trees is sufficient.

**Online publishing of a globally stable hypothesis** The next challenge we meet is to output the frequent hypothesis. The most straightforward method to do that is to repeat the idea in the batch

setting and use procedure `hist`. We can guarantee a $O(\sqrt{T})$ factor of deterioration in the privacy parameter $\epsilon$ (see Lemma 5.4) due to the repeated use of the `hist` procedure $T$ times.

Our main observation though, is that in most rounds, the frequent hypothesis does not change, allowing us to exploit the *sparse vector technique* [14], (see also [13]). The sparse vector technique is a method to answer, adaptively, a stream of queries where: whenever the answer to the query does not exceed a certain threshold the algorithm returns a negative result but *without* any cost in privacy. We pay, though, in each round where the query exceed the threshold.

We will exploit this idea in the following setting: we receive a stream of 1-sensitive lists $L_1(S), \ldots, L_T(S)$: Namely, each list $L_t$ is derived from the data $S = \{(x_1, y_1), \ldots, (x_T, y_T)\}$, and $L_t$ changes by at most one element, given a change in a single $(x_t, y_t)$. We assume that at each iteration $t$ we want to output an element $h_t \in L_t$ with high frequency. Our key assumption is that the lists are related and a very frequent element $h_t$ is also frequent at step $t + 1$. Thus in most rounds we just verify that $\mathrm{freq}_{L_t}(h_{t-1})$ is large, and only in rounds where it is too small do we use the stable histogram mechanism, paying for privacy.

Indeed, in our setting, the appearance of the frequent hypothesis may diminish by at most one each round. Once its frequency has diminished by a certain factor, then we have already made a certain fraction of the maximum possible number of mistakes. Thus, in general we only need to verify that the frequency of $h_{t-1}$ in $L_t$ is sufficiently large each round, which can be done via the sparse vector technique without loss of privacy. We next state the result more formally, the proof is provided in the full version

**Lemma 5.3.** *Consider, the procedure* $\mathrm{HistSparse}_{\eta, c, \epsilon}$ *depicted in Algorithm 2. Given a sample S, suppose Algorithm 2 receives a stream of lists, where each list is a function of S to an array of elements and each list is 1-sensitive. Then Algorithm 2 is $(\epsilon, \delta)$ differentially private and: Set*

$$\Theta_{(3)}(c, \alpha, \beta, \epsilon, \alpha) := \frac{8c(\ln T + \ln 2c/\beta)}{\alpha \epsilon}, \tag{3}$$

*and suppose:*

$$k \geq \Theta_{(4)}(c, \eta, T, \beta, \epsilon, \delta) := \max\{\Theta_{(3)}(c, \alpha, \beta, \epsilon, \alpha), \Theta_{(2)}(\eta, \beta, \epsilon, \delta)\} = \tilde{O}\left(\frac{c \ln T/\beta\delta}{\eta\epsilon}\right). \tag{4}$$

*The procedure then outputs a sequence $\{h_t\}_{t=1}^T$, where $h_t \in L_t$ such that if for each list $L_t$ there exists h such that $\mathrm{freq}_{L_t}(h) \geq \eta$ then with probability at least $(1 - 2\beta)$, for all $t \leq T$, either the algorithm aborted before step t or*

- $\mathrm{freq}_{L_t}(h_t) \geq \eta/16$.

- *If $h_{t-1} \neq h_t$:*
$$\mathrm{freq}_{L_t}(h_{t-1}) \leq \eta/8 \quad \text{and} \quad \mathrm{freq}_{L_t}(h_t) \geq \eta/4.$$

**Adaptive adversaries**   The proof for the oblivious case relies on the existence of an $h^\star$ that is consistent with the data (and independent of the random bits of the algorithm). In the adaptive case, while the sequence has to be consistent, $h^\star$ need not be determined, and the consistent hypothesis may depend on the algorithm's choices.

However, to obtain a regret bound, we rely on the standard reduction that shows that a randomized learner against oblivious adversary, can attain a similar regret against an adaptive adversary ([12], Lemma 4.1). One issue, though, is that `DP-SOA` uses random bits that are shared through time. Hence for the reduction to work we need to reinitialize the algorithm at every time-step. In this case, though, the assumptions we make for using the sparse vector technique no longer hold. Thus we can run `DP-SOA`, using `hist` (as we no longer obtain any guarantee from `HistSparse`), and we require that each output hypothesis will be $O(\epsilon/\sqrt{T}, O(\delta/T))$-DP. The privacy of the whole mechanism now follows from $T$-fold composition:

**Lemma 5.4.** *(see for example Dwork and Roth [13]) Suppose $(\epsilon', \delta')$ satisfy:*

$$\delta' = \delta/2T, \quad \text{and} \quad \epsilon' = \frac{\epsilon}{2\sqrt{2T\ln(1/\delta)}}. \tag{5}$$

*Then, the class of $(\epsilon', \delta')$-differentially private mechanisms satisfies $(\epsilon, \delta)$-differentialy privacy under T-fold adaptive composition.*

Unfortunately though, the above strategy leads to a $\sqrt{T}$ factor in the regret.

## Acknowledgments and Disclosure of Funding

The authors would like to thank Uri Stemmer for helpful discussions. N.G is supported by a Fannie & John Hertz Foundation Fellowship and an NSF Graduate Fellowship; R.L is supported by an ISF grant no. 2188/20 and by a grant from Tel Aviv University Center for AI and Data Science (TAD) in collaboration with Google, as part of the initiative of AI and DS for social good.

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
