## A  Proofs

### A.1  Proof of Theorem 4.1

**Privacy:** We begin by proving the privacy guarantees:

**Lemma A.1.** *Suppose we run Algorithm 1 with parameters $(\epsilon, \delta)$. Then the output sequence $h_1, \ldots, h_t$ is $(\epsilon, \delta)$-DP.*

**Proof.** Note that at every time step $t$, changing a single element $x_t$ changes at most one element on the list $L_t = \{A(S_{v_t^{(i)}})\}_{i=1}^{k_2}$ – specifically, the tree $i$ for which $\pi(t)$ assigns the element $x_t$. Next, note that if we fix the random bits of the algorithm, except for those that are used in the sub-procedure HistSparse (i.e. $\pi$ and the random guessing $y_v$), then each list is completely determined at step $t$ by the dataset $S$. Indeed, each $S_{v_t^{(i)}}$ is independent of $h_1, \ldots, h_T$ and the updates of the algorithm are independent of those. As such, we can think of the lists as functions of the dataset $S$.

The prerequisite assumptions for Algorithm 2 hold then (see Lemma 5.3), and by Lemma 5.3, we have that the list $h_1, \ldots, h_T$ is then $(\epsilon, \delta)$-DP. ∎

**Utility:** The core lemma behind our proof is a statement that there exists (at each iteration) a function that is frequently outputted by a fraction of the trees; the proof is deferred to Appendix A.3.

**Lemma A.2.** *Suppose $(x_1, y_1), \ldots, (x_T, y_T)$ is consistent with some hypothesis $h^\star \in \mathcal{H}$. If*

$$k_1 \geq \max\{2^{d+1}, 20\}, \quad \text{and} \quad k_2 \geq 2^{8k_1+6}k_1^2 \log \frac{5T \log k_1}{\beta} := \Theta_{(6)}(k_1, T, \beta), \qquad (6)$$

*then with probability at least $1 - \beta$, for all iterations $t \leq T$ there exists a predictor $f \neq \perp$ such that:*

$$\text{freq}_{L_t}(f) \geq \frac{2^{-4k_1}}{4k_1}.$$

We continue with the proof of Theorem 4.1, assuming Lemma A.2. The proof is an immediate corollary of the following utility lemma.

**Lemma A.3.** *Suppose Algorithm 1 is run on a sequence $(x_1, y_1), \ldots, (x_T, y_T)$, and assume that there exists $h^\star \in \mathcal{H}$ such that $h^\star(x_i) = y_i$ for all $i \in [T]$. Then, for $\beta = 1/T$, $\eta$ and $c$ as initialized in Algorithm 1, if:*

$$k_1 \geq \max\{2^{d+1}, 20\}, \quad \text{and} \quad k_2 \geq \max\{\Theta_{(4)}(c, \eta, T, \beta, \epsilon, \delta), \Theta_{(6)}(k_1, T, \beta)\} = \tilde{O}\left(\frac{2^{8 \cdot 2^d}}{\epsilon} \ln T/\delta\right).$$

*the expected number of mistakes the algorithm makes after $T$ rounds is:*

$$\mathbb{E}\left[\sum_{t=1}^{T} \mathbf{1}[h_t(x_t) \neq y_t]\right] \leq \frac{4k_1^3 \cdot 2^{2k_1} k_2}{\eta} + 1 = \tilde{O}\left(\frac{2^{8 \cdot 2^d}}{\epsilon} \ln T/\delta\right).$$

**Proof of Lemma A.3** First, setting $\beta = 1/T$ we have by assumption that $k_2 \geq \Theta_{(6)}(k_1, T, \beta)$. As such, we can turn to Lemma A.2 and setting $\eta = \frac{2^{-4k_1}}{4k_1}$ we have that, with probability $1 - 1/T$, for each list $L_t$ there is an element $f$ such that $\text{freq}_{L_t}(f) \geq \eta$. We can now apply Lemma 5.3, to obtain that, overall with probability $1 - 3/T$: either the algorithm halted, or for each $t$:

1. $\text{freq}_{L_t}(h_t) \geq \eta/16$.
2. If $h_{t-1} \neq h_t$, then

$$\text{freq}_{L_t}(h_{t-1}) \leq \eta/8 \quad \text{and} \quad \text{freq}_{L_t}(h_t) \geq \eta/4.$$

Let us denote this event by $E_0$, and we will assume for now on the $E_0$ happened.

Next, we want to show (under $E_0$) that for $c = 4k_1/\eta$, we have that

$$|\{t : \text{freq}_{L_t}(h_{t-1}) \leq \eta/8\}| \leq c.$$

To see the above, let $t$ be a time-step for which $\text{freq}_{L_t}(h_{t-1}) \leq \eta/8$, but the algorithm did not abort before time-step $t$. Set $t' < t$ be the last iteration where we called `hist` procedure (i.e. the last time we updated counter in `HistSparse`). Observe that $h_{t-1} = h_{t'}$, and note that by Item 2 we have that $\text{freq}_{L_{t'}}(h_{t'}) > \eta/4$. In particular, the Hamming distance between the lists $L_t$ and $L_{t'}$ is at least $\eta \cdot k_2/4$.

Note that for each $i \in [k_2]$, $v_t^{(i)}$ is changed between rounds $t$ and $t + 1$ only if we run the While loop in Algorithm 1 at round $t$. Note also that at each iteration of the While loop, the size of the set $\mathcal{V}_t$ is decreased by 1 (as we remove two siblings and add their parent). So $|\mathcal{V}_{t'}| - |\mathcal{V}_t| \geq \eta \cdot k_2/4$. Let $c_t$ be the number of time steps $t' \leq t$ so that $\text{freq}_{L_{t'}}(h_{t'-1}) \leq \eta/8$. At initialization we have that $|\mathcal{V}_1| = k_2 \cdot k_1$; thus, for all $t \geq 1$,

$$k_2 \cdot k_1 - \eta/4 \cdot k_2 \cdot c_t \geq 0 \Rightarrow c_t \leq 4k_1/\eta.$$

By the choice of $c = 4k_1/\eta$ in Algorithm 1, the algorithm doesn't halt and we have that, under $E_0$,

$$\forall t = 1, \ldots, T : \text{freq}_{L_t}(h_t) > \frac{\eta}{16}. \tag{7}$$

We next continue to bound the expected number of mistakes conditioned on $E_0$.

Suppose that $\pi(t)$ belongs to the $i$-th tree. Note that $\pi(t)$ is independent of $h_t$ as well as $\mathcal{V}_t$. We have, then, that with probability $1/k_1$, $\pi(t)$ is a descendent of $v_t^{(i)}$. One can observe, that for every leaf there exists a unique predecessor that belongs to $\mathcal{V}_t$. Overall then, we obtain that with probability $1/k_1$, $v_t^{(i)} = v_1$. (Recall that $v_1$ is defined in Algorithm 1 to be the unique antecedent of $\pi(t)$ that is in $\mathcal{V}_t$.)

Also, because $\text{freq}_{L_t}(h_t) > \eta/16$, with probability $\eta/16$ we have $A(S_{v_t^{(i)}}) = h_t$. Taken together we have that whenever the algorithm makes a mistake then $A(S_{v_1})$ makes a mistake with probability at least $\eta/(16k_1)$. Therefore

$$\mathbb{E}\left(\mathbf{1}[h_t(x_t) \neq y_t] \mid E_0\right) \leq \frac{16k_1}{\eta} \mathbb{E}\left(\mathbf{1}[A(S_{v_1})[x_t] \neq y_t \mid E_0]\right).$$

Again, notice that if $A(S_{v_1})$ makes a mistake, we have that $|\mathcal{V}_t|$ is reduced by at least 1. (Indeed, in this case we have that both $v_1, v_2 \in \mathcal{V}$ by choice of $v_t^{(i)}$; because we make a mistake, after adding $(x_t, y_t)$ to the sequence $S_{v_1}$, the algorithm disagrees on these two sequences, hence we run at least one iteration of the While loop that reduces the size of $\mathcal{V}_t$ by at least 1.)

As before, since at the beginning $|\mathcal{V}_1| = k_2 \cdot k_1$:

$$\mathbb{E}\left[\sum_{t=1}^{T} \mathbf{1}[h_t(x_t) \neq y_t] \mid E_0\right] \leq \frac{16k_1}{\eta} \sum_{t=1}^{T} \mathbb{E}\left(\mathbf{1}\left[A(S_{v_1})[x_t] \neq y_t \mid E_0\right]\right)$$

$$= \frac{16k_1}{\eta} \mathbb{E}\left(\sum_{t=1}^{T} \mathbf{1}\left[A(S_{v_1})[x_t] \neq y_t\right] \mid E_0\right)$$

$$\leq \frac{16k_1|\mathcal{V}_1|}{\eta}$$

$$= \frac{16k_1^2 k_2}{\eta}.$$

Hence, we obtain in expectation

$$\mathbb{E}\left[\sum_{t=1}^{T} \mathbf{1}[h_t(x_t) \neq y_t]\right] \leq \frac{k_1^2 k_2}{\eta} + \beta T \leq \frac{k_1^2 k_2}{\eta} + 3.$$

### A.2 Proof of Theorem 4.2

We consider the following procedure:

- Given $\epsilon, \delta, T$, set $\epsilon', \delta'$ as in Eq. (5).
- At each time-step $t$, run DP-SOA with privacy parameters $(\epsilon', \delta')$, $k_1, k_2$ on the input sequence $S_t = ((x_1, y_1), \ldots, (x_{t-1}, y_{t-1}))$.
- Receive a sequence $h_1^{(t)}, \ldots, h_t^{(t)}$ from DP-SOA and output $h_t = h_t^{(t)}$.

Now, we assume $k_1$ and $k_2$ are chosen so that for an oblivious sequence the conditions of Theorem 4.1 are met, and hence

- Each output $h_1^{(t)}, \ldots, h_t^{(t)}$ is $(\epsilon', \delta')$-DP w.r.t to the input sequence $S_t = ((x_1, y_1), \ldots, (x_{t-1}, y_{t-1}))$.
- For any oblivious sequence of length $T$, we have that the mistake bound is bounded by $O\left(2^{8 \cdot 2^d} / \epsilon' \ln T / \delta'\right)$.

Now, for privacy we can use Lemma 5.4. Consider the setting of privacy against an adaptive adversary as introduced in Section 3. Observe that, by our definition of the adaptive adversary, each time we apply DP-SOA, we apply it on either the sample $S_t^0 = (x_1^0, y_1^0), \ldots, (x_t^0, y_t^0)$, or $S_t^1 = (x_1^1, y_1^1), \ldots, (x_t^1, y_t^1)$, which can differ by at most one sample. Therefore, since the mechanism that outputs $h_t^{(t)}$ at step $t$ is $(\epsilon', \delta')$-DP, we obtain via Lemma 5.4 that the above adaptive online classification algorithm is $(\epsilon, \delta)$-DP.

As for utility, the result follows immediately for the standard reduction from an oblivious online learner to an adaptive one (Lemma 4.1 in [12]). Indeed, note that at step $t$ we predict $h_t$ according to a distribution $p_t$ which is completely defined by the previous sequence of examples $(x_1, y_1), \ldots, (x_{t-1}, y_{t-1})$ (it is the distribution from which the oblivious algorithm DP-SOA chooses its prediction). Thus the precondition of [12, Lemma 4.1] is verified, and we obtain the regret bound:

$$\sum_{t=1}^{T} \mathbb{E}[\mathbf{1}[h_t(x) \neq y]] \leq O\left(2^{8 \cdot 2^d} / \epsilon' \ln T / \delta'\right).$$

## A.3 Proof of Lemma A.2

Let $h^\star$ be a fixed hypothesis that is consistent with the dataset $(x_1, y_1), \ldots, (x_T, y_T)$. We will call a tree $T$ in the forest $G$ consistent if for every vertex $v$, $S_v$ is consistent with hypothesis $h^\star$ and we let $\mathcal{G}_c$ be the sub-graph that consists only of consistent trees. With these notations in mind, we now proceed to the proof. We will divide the proof into two claims; the first one, Claim A.4, gives a lower bound on the number of consistent trees.

**Claim A.4.** *For a fixed time-step* $t \leq T$, *with probability at least,* $1 - e^{-\frac{1}{2} k_2 \cdot 2^{-4 \cdot k_1}}$, *we have that* $2^{-2k_1 - 1} \cdot k_2$ *of the trees in $G$ are consistent.*

**Proof.** Note that for a tree to be consistent we only need that for every $y_{\bar{v}}$ that we guess while running the algorithm, we have that $y_{\bar{v}} = h^\star(x_{\bar{v}})$. If this happens, then all sequences $S_{\bar{v}}$ remain consistent in the tree. For each $\bar{v}$, this happens with probability $1/2$, independent on the sequence and the other labels $y_{\bar{v}}$. Hence each tree is consistent with probability at least $2^{-2 \cdot k_1}$ (the number of vertices) and this is independent of the other trees. Thus, applying the Chernoff bound, we obtain that if $M_t$ is the number of conistent trees at time $t$, then:

$$\mathbb{P}\left(M_t \leq (2^{-2k_1} - 2^{-(2k_1+1)}) \cdot k_2\right) \leq e^{-2k_2 \cdot 2^{-2(2 \cdot k_1 + 1)}} \tag{8}$$

■

The next step is to prove that (with high probability) there exists a function $f$ that appears frequently in the list $\{A(S_v)\}$ of vertices that belong to consistent trees, which we do next.

First let us denote by $\Xi = (\pi, \{y_v\}_{v \in V})$ the random seed, or internal bits, of DP-SOA, not including the random bits of the mechanisms HistSparse. Note that, at each time-step, the sets $S_v$, and $\mathcal{V}_t$ are completely determined by $\Xi$ (and the oblivious sequence). In particular, the state of the forest is completely independent of the output hypotheses picked by HistSparse.

Let $\mathcal{G}_c(\Xi; t)$ denote the subgraph of consistent trees given $\Xi$ at time $t$ and let $F_k(\Xi; t)$ be the multiset that consists of all labeled subtrees (at time step $t$) of consistent trees whose root is a depth-$k$ vertex. We will often, with slight abuse of notation, associate a tree in $F_k(\Xi; t)$ to its $S_v$-labeled root $v$, which is a depth-$k$ vertex of some consistent tree; thus we will write, at times, "for each $v$ in $F_k(\Xi; t)$". (Also note that it may be the case that for some depth-$k$ vertices $v$, $S_v = \emptyset$; the subtrees rooted at such $v$ are still included in $F_k(\Xi; t)$). Also, let us say that the (multi)set $F_k$ is $f$-*heavy* if, for at least $2^{-k_1}|F_k|$ of the vertices $v$ in $F_k$ we have that $f = A(S_v) \neq \perp$.

Then we have the following claim:

**Claim A.5.** *For a fixed time-step* $t \leq T$, *let* $\mathcal{F}$ *denote the event that for some* $k \leq \log k_1 + 1$ *and* $f$, $F_k$ *is* $f$-*heavy. then,*

$$\mathbb{P}(\mathcal{F}) \geq 1 - 2 \log k_1 \cdot e^{-\frac{2^{-4k_1 - 1}}{9} k_2}. \tag{9}$$

**Proof.** The crucial observation is that, because the distribution of $\pi$ is invariant under permutation of the leaves, then given $F_k$ and $\mathcal{G}_c$, the distribution $\pi$ of the assignments of data points can be viewed as randomly sampling (without replacement) elements from $F_k$ and assigning to each subtree its appropriate depth-$k$ vertex as a root.

Specifically, let us say that a vertex $v$ is *active* if it belongs to a consistent tree. Now, let $V_k(\Xi; t)$ be the set of labeled depth-$k$ active vertices which are right-children of their parents. For each $v \in V_k(\Xi; t)$, denote by $X_v$ the random variable defined as follows: $X_v = 1$ if $A(S_v) = A(S_{s(v)})$ and $v, v_{s(v)} \in \mathcal{V}_t$ at the end of the While loop at step $t$ of Algorithm 1, and $X_v = 0$ otherwise (recall that $t$ is fixed). And further, denote

$$E(\Xi; k) = \mathbb{E}\left[\frac{1}{|V_k|} \sum_{v \in V_k} X_v \;\middle|\; F_k(\Xi; t), \mathcal{G}_c(\Xi; t)\right].$$

We claim the following bound holds for the time-step $t$:

$$\Pr_{\Xi}\left(\max_k\left\{E(\Xi; k) - \frac{1}{|V_k|} \sum_{v \in V_k} X_v\right\} > 2^{-k_1}\right) \leq \log k_1 e^{-\frac{2^{-2k_1}|V_k|}{18}}. \tag{10}$$

To establish Eq. (10), note that for a fixed $k \leq \log k_1$ and a set $F_k$, by symmetry of the distribution of $\pi$, the joint distribution of all $X_v$ does not change if we resample the labels $S_v$ for all vertices $v$ in $F_k$, from this set of all labels, without replacement. Note that changing a single element $S_v$ will change at most one random variable $X_v$, and as such we get that $\frac{1}{|V_k|} \sum X_v$ is $\frac{1}{|V_k|}$-sensitive. Since we randomly draw $2|V_k|$ elements, we can thus use Lemma 5.2 to obtain that for a fixed $k$, $F_k$ and $\mathcal{G}_c$:

$$\Pr_{\Xi} \left( E(\Xi; k) - \frac{1}{|V_k|} \sum_{v \in V_k} X_v > 2^{-k_1} \mid F_k(\Xi; t) = F_k, \mathcal{G}_c(\Xi; t) = \mathcal{G}_c \right)$$

$$= \Pr_{\Xi} \left( \mathbb{E} \left[ \frac{1}{|V_k|} \sum_{v \in V_k} X_v \mid F_k, \mathcal{G}_c \right] - \frac{1}{|V_k|} \sum_{v \in V_k} X_v > 2^{-k_1} \mid F_k, \mathcal{G}_c \right)$$

$$\leq e^{-\frac{2^{-2k_1}|V_k|}{18}}.$$

Eq. (10) now follows by taking expectation over $F_k, \mathcal{G}_c$ as well as a union bound over the $\log k_1$ possible values of $k \leq \log k_1$.

We next observe that for any consistent tree there exists a vertex $v$, such that $v, s(v) \in \mathcal{V}_t$ and $A(S_v) = A(S_{s(v)})$. Indeed, if this is not the case, then one can prove by induction that the tree's root $v_r$ is in $\mathcal{V}$. However, the sequence $S_{v_r}$ makes $\log k_1 \geq d + 1$ mistakes, which is a contradiction to the consistency of the tree. Then, what we showed so far is that in any consistent tree there exists $v$ such that $X_v = 1$. Thus, applying pigeon-hole principle, we obtain that for any $\pi$ there exists a $k \leq \log k_1$ such that

$$\frac{1}{|V_k|} \sum_{v \in V_k} X_v \geq \frac{1}{k_1 \log k_1} \geq 2^{-k_1+1}.$$

Together with Eq. (10) we get that, given $\mathcal{G}_c$, with probability at least $1 - \log k_1 \cdot e^{-\frac{2^{-2k_1+1}|V_k|}{18}}$, for some $k$ we have that

$$E(\Xi; k) > 2^{-k_1}.$$

Finally, (where for ease of notation we neglect the dependence of $F_k, \mathcal{G}_c$ in $\Xi$) we have

$$E(\Xi; k) = \frac{1}{|V_k|} \sum_{v \in V_k} \mathbb{P} \left( A(S_v) = A(S_{s(v)}) \neq \perp \mid F_k, \mathcal{G}_c \right)$$

$$= \frac{1}{|V_k|} \sum_{v \in V_k} \sum_{f \neq \perp} \mathbb{P}(A(S_v) = f \mid F_k, \mathcal{G}_c) \, \mathbb{P} \left( A(S_{s(v)}) = f \mid A(S_v) = f, F_k, \mathcal{G}_c \right)$$

$$\leq \frac{1}{|V_k|} \sum_{v \in V_k} \sum_{f \neq \perp} \mathbb{P}(A(S_v) = f \mid F_k, \mathcal{G}_c) \, \mathbb{P} \left( A(S_{s(v)}) = f \mid F_k, \mathcal{G}_c \right)$$

$$= \frac{1}{|V_k|} \sum_{v \in V_k} \sum_{f \neq \perp} (\mathbb{P}(A(S_v) = f \mid F_k, \mathcal{G}_c))^2$$

$$\leq \frac{1}{|V_k|} \sum_{v \in V_k} \max_{f \neq \perp} \mathbb{P}(A(S_v) = f \mid F_k, \mathcal{G}_c)$$

$$= \max_{f \neq \perp} \mathbb{P}(A(S_{v_0}) = f \mid F_k, \mathcal{G}_c),$$

where the first inequality follows from the fact that $S_v$ are sampled without replacement, hence the distribution for $A(S_{s(v)}) = f$ given that we already sampled such an element reduces. The last equality follows from the fact that the distribution of $S_v$, conditioned on $F_k, \mathcal{G}_c$, is identical for all $v \in V_k$; in the last line we set $v_0$ to be an arbitrary vertex in $V_k$.

Finally, using Claim A.4, and noting that $|V_k|$ is at least the number of consistent trees, we have that with probability

$$1 - \log k_1 \cdot e^{-\frac{2^{-2k_1+1}|V_k|}{18}} - e^{-k_2 \cdot 2^{-4 \cdot k_1 - 1}} \geq 1 - 2 \log k_1 \cdot e^{-\frac{2^{-4k_1-1} k_2}{9}},$$

for some $k$, we have
$$\max_{f \neq \perp} \mathbb{P}(A(S_{v_0}) = f|F_k, \mathcal{G}_c) \geq 2^{-k_1},$$
where again $v_0$ is an arbitrary vertex in $V_k$. Since $S_{v_0}$ is sampled uniformly at random from the set of $S_v$ for $v \in F_k(\Xi; t)$, the left-hand side of the above inequality is simply the fraction of $S_v$, for $v \in F_k(\Xi; t)$ for which $A(S_v) = f$. In particular, we obtain that $F_k(\Xi; t)$ is heavy. ∎

The final claim we will need bounds the number of times we have $A(S_v) = A(S_{s(v)}) = f$ given the $F_k$ is heavy:

**Claim A.6.** *For a fixed time-step $t \leq T$, recall that $\mathcal{F}$ is the event that $F_k$ is $f$-heavy for some $f$ and $k$. Let $E$ be the event that for at least $2^{-2k_1-1}k_2$ of the trees, there exists a vertex $v$ such that $A(S_v) = A(S_{s(v)}) = f$, then if $k_1 \geq 20$:*

$$\mathbb{P}(E|\mathcal{F}) \geq 1 - 2e^{-\frac{2^{-8k_1-6}}{9}k_2}. \tag{11}$$

**Proof.** Fix the set of consistent trees $\mathcal{G}_c$, and assume that the number of consistent trees is at least $2^{-k_1-1} \cdot k_2$. We can assume that $k_2 \geq 2^{2k_1+2}$ (otherwise, since $k_1 \geq 20$ the bound is trivial), hence $|F_k| \geq 2^{k_1+1}$, for any $k$ (as $|F_k|$ is bounded below by the number of consistent trees).

Let us condition $\pi$ on the consistent trees $\mathcal{G}_c$ and $F_k$, which we will assume to be $f$-heavy. Again, we use the fact that conditioned on $F_k, \mathcal{G}_c$, the joint distribution of all $S_v$ ($v \in F_k$) is unchanged if we randomly resample each $S_v$-labeled vertex $v$ from $F_k$, without replacement. In particular we have that, for any $k$-depth vertex $v$:

$$\begin{aligned}
\mathbb{P}(A(S_{s(v)}) = f|A(S_v) = f, F_k, \mathcal{G}_c) &\geq 2^{-k_1} - \frac{1}{|F_k|} \\
&\geq 2^{-k_1} - 2^{-k_1-1} \qquad |F_k| \geq 2^{k_1+1} \\
&= 2^{-k_1-1}.
\end{aligned}$$

For $i \in [k_2]$, we now set $X_i$ to be the random variable defined by: $X_i = 1$ if there exists $v$ in the $i$-th tree such that $A(S_{s(v)}) = A(S_v) = f$, and $X_i = 0$ otherwise. For each consistent tree $i$, and for any depth-$k$ vertex $v$ of tree $i$, using the fact that $F_k$ is $f$-heavy, we have:

$$\begin{aligned}
\mathbb{E}[X_i|F_k, \mathcal{G}_c] &\geq \mathbb{P}(A(S_v) = A(S_{s(v)}) = f|F_k, \mathcal{G}_c) \\
&= \mathbb{P}(A(S_v) = f \mid F_k, \mathcal{G}_c) \cdot \mathbb{P}(A(S_s(v) = f \mid A(S_{s(v)}) = f, F_k, \mathcal{G}_c) \\
&\geq 2^{-k_1} \mathbb{P}(A(S_s v) = f|A(S_v) = f, F_k, \mathcal{G}_c) \\
&\geq 2^{-k_1} \cdot 2^{-k_1-1} \\
&\geq 2^{-2k_1-1}.
\end{aligned}$$

So if $2^{-2k_1-1} \cdot k_2$ of the trees are in $\mathcal{G}_c$, i.e. are consistent, we have that

$$\mathbb{E}\left[\frac{1}{k_2}\sum_{i=1}^{k_2} X_i \mid F_k, \mathcal{G}_c\right] \geq 2^{-2k_1-1} \cdot 2^{-2k_1-1} = 2^{-4k_1-2}. \tag{12}$$

We again exploit the fact that changing the label $S_v$ of a single vertex $v$ in a tree changes at most one random variable $X_i$, and use Lemma 5.2 to obtain a high probability rate. In particular, for any set of consistent trees $\mathcal{G}_c$ that includes $2^{-2k_1-1} \cdot k_2$ of the trees, and for any heavy $F_k$:

$$\mathbb{P}\left(\frac{1}{k_2}\sum_{i=1}^{k_2} X_i \leq 2^{-4k_1-3} \mid F_k, \mathcal{G}_c\right) \leq e^{\frac{2^{-8k_1-6}}{9}k_2}. \tag{13}$$

Finally, we take expectation over heavy $F_k$. Note that $F_k$ determines if $F_j$ is heavy for all $j \leq k$, meaning that we may take the expectation of Eq. (13) over only those $F_k$ for which the determined $F_j$ is not heavy for all $j < k$. And by Claim A.4, $\mathcal{G}_c$ consists of $2^{-2k_1-1} \cdot k_2$ of the trees with probability at least $1 - e^{-k_2 \cdot 2^{-4k_1-2}}$. Hence

$$\mathbb{P}(E|\mathcal{F}) \geq 1 - e^{-\frac{2^{-8k_1-6}}{9}k_2} - e^{-2^{-4k_1-2}k_2} \geq 1 - 2e^{-\frac{2^{-8k_1-6}}{9}k_2}.$$

∎

**Concluding the proof of Lemma A.2** We are now ready to conclude the proof of Lemma A.2. First note that if a vertex satisfies $A(S_v) = A(S_{s(v)}) \neq \perp$ then we must have $v \in \mathcal{V}_t$. Indeed, since for both $S_v, S_{s(v)} \neq \perp$, they must at some point have been in $\mathcal{V}$ (because every time we initialize $S_v$ we also add $v$ to $\mathcal{V}$). And whenever we take $v$ out of $\mathcal{V}$ then we must also take $s(v)$, but we take them out only if $A(S_v) \neq A(S_{s(v)})$).

As such, for any fixed $f$, for any tree that contains a vertex $v$ such that $A(S_v) = A(S_{s(v)}) = f$, with probability at least $1/k_1$ we have that $A(v_t^{(i)}) = f$ (as $v_t^{(i)}$ is chosen randomly, at each time-step the tree is updated). Now utlizing Claims A.5 and A.6 we obtain that with probability at least

$$1 - 2e^{-\frac{2^{-8k_1-6}}{9}k_2} - 2\log k_1 e^{-\frac{2^{-4k_1-1}}{9}k_2} \geq 1 - 4\log k_1 e^{-\frac{2^{-8k_1-6}}{9}k_2},$$

at least $2^{-2k_1-1}k_2$ of the trees contain a vertex $v$ such that $A(S_v) = A(S_{s(v)}) = f$ for some $f \neq \perp$ (independent of the tree).

By the Chernoff bound, we obtain that for at least $\frac{2^{-4k_1-2}k_2}{k_1}$ of these trees $i$, we choose $v_t^{(i)}$ satisfying $A(v_t^{(i)}) = f$, with probability at least $1 - e^{-\frac{2^{-8k_1-4}}{k_1^2}\cdot k_2}$.

To conclude, for any fixed $t$, with probability at least

$$1 - e^{-\frac{2^{-8k_1-4}}{k_1^2}k_2} - 4\log k_1 e^{-\frac{2^{-8k_1-6}}{9}k_2} \geq 1 - 5\log k_1 e^{\frac{2^{-8k_1-6}}{k_1^2}k_2},$$

for $\frac{2^{-4k_1-2}}{k_1}$ fraction of the trees $i$ we have $A(S_{v_t^{(i)}}) = f$ for some fixed $f$. The result now follows from a union bound over $t \leq T$.

## A.4 Proof of Lemma 5.3

**Privacy** For privacy, the proof is verbatim the proof that `sparse` is private provided in [13] (but instead of publishing the answer to a linear query everytime a threshold is passed, we output a frequent hypothesis). First, we consider the following variant of the procedure `Above-threshold` introduced in [13]:

**Theorem A.7** ([13], Thm 3.26). *There exists a $(\epsilon, 0)$-DP procedure, `Above-threshold`$_{\theta,c,\epsilon}$ (depicted in Algorithm 3, that receives an adaptive sequence of queries $Q_1, \ldots, Q_T$ that are $1/k$ sensitive and outputs a list $\{a_t\}_{t=1}^T$ such that if:*

$$k \geq \Theta_{(3)}(c, \alpha, \beta, \epsilon, \alpha) := \frac{8c(\ln T + \ln 2c/\beta)}{\alpha\epsilon}, \tag{14}$$

*then for any sequence $Q_1, \ldots, Q_T$ such that $|\{t : Q_t(D) \geq \theta - \alpha\}| \leq c$, with probability $1 - \beta$:*

- *For all $a_i = \top$: $Q_i(D) \geq \theta - \alpha$.*

- *For all $a_i = \perp$: $Q_i(D) \leq \theta + \alpha$.*

---

**Algorithm 3** `Above-threshold`

---

    **Initialize:** parameters $\epsilon, \theta, c$.

    Let $\sigma = 2c/(k\epsilon)$

    Let $\theta_0 = \theta + \mathrm{LAP}(\sigma)$.

    Let counter $= 0$

    **for** each list $L_t$ **do**

      Receive a $1/k$ sensitive query $Q_t(D)$

      Let $v_i = \mathrm{LAP}(2\sigma)$

      **if** $Q_t(D) + v_i \geq \theta$ **then**

        output $\top$.

        Set counter = counter $+ 1$.

        Let $\theta_{\mathrm{counter}} = \theta + \mathrm{LAP}(\sigma)$

      **else**

        output $\bot$

      **end if**

      **if** counter $\geq c$ **then**

        ABORT

      **end if**

    **end for**

---

We observe that Algorithm 2 is the adaptive composition of `Above-threshold`, together with the `hist` mechanism with parameters $(\epsilon'/(2c), \delta'/(2c))$. Moreover since each list changes by at most one element if we change a single point in the database, we have that the queries $Q_t(D) = 1 - \mathrm{freq}_{L_t}(h_{t-1})$ are $1/k$ sensitive. Hence by standard composition we obtain that the algorithm is $(\epsilon, \delta)$-DP.

**Utility** As for accuracy, first note that at each round $t$ we choose as a query

$$Q_t(L_t) = 1 - \mathrm{freq}_{L_t}(h_{t-1}).$$

By our choice of parameters (and standard union bound), we have that with probability $(1 - 2\beta)$ the following happens at each round: Whenever the algorithm chooses $h_t = h_{t-1}$ we have that:

$$1 - \mathrm{freq}_{L_t}(h_t) = 1 - \mathrm{freq}_{L_t}(h_{t-1}) = Q_t(D) \leq \theta + \eta/32 = 1 - \eta/16 \Rightarrow \mathrm{freq}_{L_t}(h_t) \geq \eta/16,$$

and at each round that the algorithm calls *hist* we have by the guarantee of *hist* that:

$$\mathrm{freq}_{L_t}(h_t) \geq \eta/4,$$

and moreover

$$1 - \mathrm{freq}_{L_t}(h_{t-1}) = Q_t(L_t) \geq \theta - \eta/32 = 1 - \eta/8 \Rightarrow \mathrm{freq}_{L_t}(h_{t-1}) \leq \eta/8.$$

### A.5 Proof of Lemma 5.2

The main observation is that if we let $(i, j)$ be the permutation that switches between $i$ and $j$, a uniform randomly chosen permutation can be written as

$$\pi = (N, a_N) \circ ((N-1), a_{N-1}) \circ \ldots \circ (3, a_3) \circ (2, a_2),$$

where each $a_i$ is an independent random variable distributed uniformly on the set $\{1, \ldots, i\}$. An equivalent way to generate $n$ random variables $\bar{Z} = (Z_1, \ldots, Z_n)$ sampled without replacement from $\mathcal{Z} = \{z^{(1)}, \ldots, z^{(N)}\}$ is as follows: first choose a permutation $\pi$ uniformly at random, then set $(i_1, \ldots, i_n) = (\pi(N), \ldots, \pi(N - n + 1))$, and finally set $(Z_1, \ldots, Z_n) = (z^{(i_1)}, \ldots, z^{(i_n)})$. In particular, the random variable $\bar{Z} = (Z_1, \ldots, Z_n)$ is completely determined by the independent random variables $a_N, \ldots, a_{N-n+1}$. Let us write this mapping from $a_N, \ldots, a_{N-n+1}$ to $Z_1, \ldots, Z_n$ as $(Z_1, \ldots, Z_n) = G(a_N, \ldots, a_{N-n+1})$. Also note that changing a single variable $a_i$ changes at most the position of 3 elements of $G(a_N, \ldots, a_{N-n+1})$. Hence, via the triangle inequality, we obtain that, for any tuples $\bar{a} = (a_N, \ldots, a_{N-n+1})$ and $\bar{a}' = (a'_N, \ldots, a'_{N-n+1})$ that are of Hamming distance at most 1,

$$|F(G(\bar{a})) - F(G(\bar{a}'))| \leq 3c.$$

Thus, considering $F \circ G$ as a function of $a_N, \ldots, a_{N-n+1}$, we obtain the desired result via the standard Mcdiarmid's inequality.