# OpenReview forum: "Littlestone Classes are Privately Online Learnable"
_NeurIPS.cc/2021/Conference — NeurIPS 2021 Spotlight_

### Official Review · Reviewer_YBsr · 2021-07-15

**Rating:** 7
**Confidence:** 3

**Summary:**

The paper considers the problem of online learning hypothesis class $H$ under an additional privacy constraint: The sequence of hypotheses $h_1, h_2, \ldots, h_T \in H$ chosen by the learner must be differentially private w.r.t. the data sequence $(x_1, y_1)$ through $(x_T, y_T)$.

It is well-known that a hypothesis class $H$ is online-learnable if and only if its Littlestone dimension $d$ is finite. A sequence of recent breakthrough proved that $d < +\infty$ is also sufficient and necessary for $H$ to be privately learnable (in the offline/batch setting). This work makes a further and stronger connection: online learnability is equivalent to private online learnability (albeit this is only proved for the realizable setting).

Assuming the realizable setting, the following quantitative results are given:
- When $H$ has Littlestone dimension $d$ and the adversary is oblivious, $\tilde O(2^{O(2^d)}\log T)$ regret is possible. In particular, for $d = O(1)$, this is comparable to the non-private setting (in which $O(d)$ regret is achievable) up to a log factor.
- For adaptive adversaries, a modification of the algorithm gives a regret bound of $\tilde O(2^{O(2^d)}\log \sqrt{T})$.

The analysis follows the following steps:
- Achieve an online version of "global stability", meaning that some hypothesis $h^* \in H$ is returned with an $\Omega(1)$ probability. This step is roughly done by randomly allocating the data points to the leaves in a forest structure, and then propagate the data upwards whenever two sibling nodes disagree.
- In the batch setting, the analysis of the above scheme relies on the i.i.d. property, which no longer holds for the online setting. Nevertheless, the authors showed that the random allocation is sufficient to achieve a similar guarantee, using a version of Mcdiarmid's inequality.
- Finally, in the batch setting we can publish a model by running a private version of heavy hitters. To publish $T$ models in the online setting, however, leads to severe ($\sqrt{T}$) degradation of the privacy parameters. To address this issue, it is noted that the frequent hypothesis tends to be stable over time, which enables the use of the sparse vector technique.

**Limitations And Societal Impact:**

Adequately addressed.

**Main Review:**

The authors gave a clear introduction of the setting of interest, background on online learning and differential privacy, and the results are clearly stated. On the other hand, I found the technical sections (5 and 6) harder to follow as the proofs are quite involved. That said, I think the authors in general did a good job in presenting the algorithm, sketching the analyses, and explaining why the adaptive setting becomes more challenging.

This work adds on to a series of exciting recent results on the private learnability of binary hypothesis classes. While being built on some ideas/tools from previous work, both the algorithm and its analysis are nontrivial. Although the paper leaves a few important questions open (e.g., improving the dependence on both $d$ and $T$, and extending to the more challenging agnostic case), I found the current results quite solid and should be published. Therefore, I recommend acceptance.

Minor comments:
- Line 99: "(private )"
- Line 206: Should all $(x_v, y_v)$ on this line be $(x_{\overline{v}}, y_{\overline{v}})$ instead?

===Added After Author Response===

I have read the authors' response as well as the other reviews. My overall evaluation of the paper remains the same.

**Time Spent Reviewing:**

2.5

---

> ### Author Response · Authors · 2021-08-08
> **Thanks!**
>
> Thanks for the positive feedback. We will make serious attempt to improve sections 5 and 6.
>
> Regarding line 206: Yes, thanks, all $(x_v, y_v)$ on that line should be $(x_{\bar v}, y_{\bar v})$.

---

### Official Review · Reviewer_pd6C · 2021-07-16

**Rating:** 8
**Confidence:** 3

**Summary:**

The paper presents an on line classification algorithm which satisfies the differential privacy property. In the realizable setting,  if the Littlestone dimension of the hyothesis space is constant, against oblivious adversary the regret (mistake bound) is equivalent (up to logarithmic factor) to that achieved in the non-differential privacy setting. The loss is in general significant (doubly exponential) in terms of Littlestone dimension. For constant Littlestone dimension,  and against adaptive adversary, the regret obtained is sublinear but grows to order of square root of the time horizon T.

**Limitations And Societal Impact:**

The paper sufficiently discusses the limitations and does not present potential negative societal impact.

**Main Review:**

Strength: the work strengthens recent results connecting differential privacy and on line learnability by showing scenarios where on line learning can be directly endowed with differential privacy property.
Weakness: the regret grown doubly exponentially with the Littlestone dimension of the hypotheses class.
Originality: the main proof tool bears similarity with recent analogous results for the stochastic setting, however the authors have to overcome some specific new problems to adapt it to the on line setting.
Overall, I think the results are solid and consistute a significant contribution.
Clarity and presentation: The main algorithmic elements and proof techniques are well explained and the paper is easy to read. However, there are several syntactic errors (words and parts of sentences are swapped, prepositions are missing, verbs are not properly inflected, etc.) and the paper would benefit from a thorough proofreading

**Time Spent Reviewing:**

8

---

> ### Author Response · Authors · 2021-08-08
> **Thanks!**
>
> Thank you for your positive feedback and comments. Indeed, improving the double-exponential rate will strengthen our result, and we consider it as a future work.
>
> The paper will go a thorough proofreading, and syntactic errors will be taken care of.

---

### Official Review · Reviewer_aXPb · 2021-07-16

**Rating:** 7
**Confidence:** 3

**Summary:**

The paper introduces the problem of online classification under a differential privacy constraint. There is a learner and an adversary. Specifically, it is required that the sequence of hypotheses outputted by the learner is (epsilon,delta)-differentially private in the sequence of examples presented by the adversary.

The paper focuses on the realizable setting, where the sequence of examples chosen by the adversary is consistent with some unknown hypothesis that lies in a hypothesis class known to both the learner and the adversary.

Two cases are studied. The adversary can be oblivious or adaptive. For oblivious adversaries, the paper proposes an algorithm with regret that is doubly exponential in Littlestone dimension and log(T) factor. For adaptive adversaries, an upper bound doubly exponential in Littlestone dimension times sqrt{T} regret is achieved.


**Limitations And Societal Impact:**

In my opinion, the authors have adequately addressed the limitations of their work. I would encourage the authors to discuss societal impacts of their work even if it’s currently of theoretical nature.



**Main Review:**

I think this is a solid contribution to the area of differentially-private learning. While quantitatively the bounds are not great, e.g. the doubly-exponential dependence on the Littlestone dimension, I think formulating the problem and proving the first non-trivial upper bounds is substantial.

The paper is very well-written and easy to follow.

Technically, the paper relies on the property of global stability introduced in prior work. The authors show how to construct an online version of a globally stable algorithm to obtain the log(T) factor in the oblivious case with a novel application of the sparse vector technique. The adaptive case is achieved via a reduction to the oblivious case and using T-fold composition, but this incurs a sqrt{T} factor.

It might be helpful for readers of Section 6 if the authors could separate how an online version of a globally stable learner implies the online private learner, and how they construct this globally stable learner from SOA.

I would appreciate it if the authors could clarify the following. The authors considered requiring the sequence of hypotheses h_1,...,h_T to be differentially-private, but what about requiring only the sequence of predictions h_1(x_1),...,h_T(x_T) to be differentially-private? Is this notion of privacy different? is it useful?

In Theorem 4.1 and Theorem 4.2, the upper bounds on the mistake bound of the proposed algorithms incur some dependence on T (log(T) and sqrt{T}). We know that in standard online classification, in the realizable setting, no such dependence is incurred. Can the authors comment on whether such dependence on T is necessary in private online classification?

Can the authors clarify if their algorithm can be made to work with any black-box online learner? Is knowledge of the class H required or is having a black-box online learner sufficient?

Similar to online-to-batch conversions in standard classification, can private online learners be leveraged to construct private learners in the stochastic setting? This could be further motivation for studying the problem of private online learning.

Minor comments.
- Line 128: event E \subseteq W?



**Time Spent Reviewing:**

5

---

> ### Author Response · Authors · 2021-08-08
> **Thanks!**
>
> Thank you for the positive feedback!
>
> > I would appreciate it if the authors could clarify the following. The authors considered requiring the sequence of hypotheses h_1,...,h_T to be differentially-private, but what about requiring only the sequence of predictions h_1(x_1),...,h_T(x_T) to be differentially-private? Is this notion of privacy different? is it useful?
>
> This is a very good point, and we should add a discussion and compare the models. The models are indeed different and the guarantee in terms of privacy is weaker, but, potentially, might allow improved rates. For instance, in the iid setting, it was studied in [1] below, where sample complexity linear in the VC dimension was shown to be achievable.
>
> [1] Amos Beimel, Kobbi Nissim, Uri Stemmer. "Private Learning and Sanitization: Pure vs. Approximate Differential Privacy", 2014.
>
> > Can the authors comment on whether such dependence on T is necessary in private online classification?
>
>  We are unaware of any lower bounds for the dependence on T, this is a good question and we will add a discussion.
>
> > Can the authors clarify if their algorithm can be made to work with any black-box online learner? Is knowledge of the class H required or is having a black-box online learner sufficient?
>
> The black-box online learner A used in DP-SOA can be any online learning algorithm which achieves the optimal mistake bound (in particular, it must change its internal state after making a mistake). Explicit knowledge of H is not required, though explicit knowledge of the hypotheses the algorithm A produces is required (e.g., to choose an x_{\bar v} in the 2nd line of the While loop of DP-SOA).
>
> > Can private online learners be leveraged to construct private learners in the stochastic setting?
>
> Yes, one can establish online-to-batch conversions for the setting of private learning (but currently this will not yield improved rates and will not even compare to the sample-efficient known rates for the stochastic setting).
>
> >Line 128:
>
> Yes, thanks, it should be $E \subset W$.

---

### Official Review · Reviewer_wJX6 · 2021-07-24

**Rating:** 8
**Confidence:** 4

**Summary:**

The submission considers the problem of online classification, in which a learner is provided with a stream of labeled examples and must return a hypothesis at each iteration. In particular, the authors look at online learning in a differentially private setting for both the case of oblivious adversaries as well as that of adaptive adversaries (in the realizable setting) and prove mistake bounds.

**Main Review:**

The paper is a strong submission and makes progress on an important question. Private learnability in the offline setting has been well-studied in recent years, with the results of Alon et al, Bun et al., and Ghazi et al. that have established the connection between finite Littlestone dimension and private learnability. How such a result extends to private *online* learnability is a natural question, and some works have studied regret bounds for private online convex optimization. This submission considers arbitrary hypothesis classes of finite Littlestone dimension and shows that a hypothesis class with Littlestone dimension d can be learned privately with a mistake bound that scales as 2^(O(2^d)).

The paper is fairly easy to follow, although some more inuition could improve the readability of Algorithm 1. There are some minor typos (e.g., the word "adversarial" was misspelled numerous times on page 3).

In terms of novelty: The techniques are based on ideas that have been explored in other works. The DP-SOA algorithm, for instances, relies on using copies of a non-private SOA algorithm on different subsequences. Moreover, the mistake bounds proceed via construction of an online version of a globally stable algorithm, which draws upon ideas from Bun et al. However, the way in which these ideas are tied together is nontrivial, which merits originality.

The paper appears to be technically sound, although I have not verified all the details of the proofs (e.g., material in the supplementary material).

**Time Spent Reviewing:**

4

---

> ### Author Response · Authors · 2021-08-08
> **Thanks!**
>
> We thank the reviewer for the comments and suggestions. We will fix typos as well as try to clarify algorithm 1.

---

### Decision · Program_Chairs · 2021-09-27

**Decision:**

Accept (Spotlight)

**Comment:**

A very nice paper that does what it title says: gives a differentially-private online algorithm for classes of finite Littlestone dimension.  The paper is clearly written and gives a good background of the problem, and while the paper mostly makes use of known techniques, it still gives a nontrivial and interesting result.